# Bone-Regeneration Therapy Using Biodegradable Scaffolds: Calcium Phosphate Bioceramics and Biodegradable Polymers

**DOI:** 10.3390/bioengineering11020180

**Published:** 2024-02-13

**Authors:** Kaoru Aoki, Hirokazu Ideta, Yukiko Komatsu, Atsushi Tanaka, Munehisa Kito, Masanori Okamoto, Jun Takahashi, Shuichiro Suzuki, Naoto Saito

**Affiliations:** 1Physical Therapy Division, School of Health Sciences, Shinshu University, Matsumoto 390-8621, Japan; 2Department of Orthopaedic Surgery, Shinshu University School of Medicine, Matsumoto 390-8621, Japan; ideta@shinshu-u.ac.jp (H.I.); yukikomatsu25@shinshu-u.ac.jp (Y.K.); tanaatsu@shinshu-u.ac.jp (A.T.); mune0527@yahoo.co.jp (M.K.); ryouyuma@shinshu-u.ac.jp (M.O.); jtaka@shinshu-u.ac.jp (J.T.); 3Department of Orthopaedic Surgery, Matsumoto Medical Center, Matsumoto 390-8621, Japan; shuichiro-suzuki@live.jp; 4Institute for Biomedical Sciences, Interdisciplinary Cluster for Cutting Edge Research, Shinshu University, Matsumoto 390-8621, Japan; saitoko@shinshu-u.ac.jp

**Keywords:** bone defect, bone regeneration, scaffold, calcium phosphate, biodegradable polymer

## Abstract

Calcium phosphate-based synthetic bone is broadly used for the clinical treatment of bone defects caused by trauma and bone tumors. Synthetic bone is easy to use; however, its effects depend on the size and location of the bone defect. Many alternative treatment options are available, such as joint arthroplasty, autologous bone grafting, and allogeneic bone grafting. Although various biodegradable polymers are also being developed as synthetic bone material in scaffolds for regenerative medicine, the clinical application of commercial synthetic bone products with comparable performance to that of calcium phosphate bioceramics have yet to be realized. This review discusses the status quo of bone-regeneration therapy using artificial bone composed of calcium phosphate bioceramics such as β-tricalcium phosphate (βTCP), carbonate apatite, and hydroxyapatite (HA), in addition to the recent use of calcium phosphate bioceramics, biodegradable polymers, and their composites. New research has introduced potential materials such as octacalcium phosphate (OCP), biologically derived polymers, and synthetic biodegradable polymers. The performance of artificial bone is intricately related to conditions such as the intrinsic material, degradability, composite materials, manufacturing method, structure, and signaling molecules such as growth factors and cells. The development of new scaffold materials may offer more efficient bone regeneration.

## 1. Introduction

The discovery of pluripotent cells such as induced pluripotent stem cells (iPS cells) has led to extensive research in regenerative medicine [1,2]. However, organs do not comprise of only one type of cell. For example, in the case of the heart, tissues such as myocardia, blood vessels, cardiac membranes, nerves, valves, and tendinous cords exhibit complex structures and functions that present many obstacles before heart regeneration can be applied to clinical practice. Regenerative medicine cannot simply be realized by increasing the number of cells in the target organ or tissue, but requires the proliferation of these cells to form organs and tissues, for which scaffolding plays an important role [3].

When a bone defect occurs due to trauma or treatment of a bone tumor, there are various treatment options that include joint arthroplasty, autologous bone grafting, allogeneic bone grafting, synthetic bone grafting, and the induced membrane technique [4,5,6,7,8,9,10,11,12,13,14,15]. The appropriate treatment strategy depends on the size and location of the bone defect and the type of bone [16].

Joint arthroplasty such as endoprosthetic replacement surgery is performed when the bone defect is relatively large or occurs near the joint. Osteosarcoma is the most common malignant bone tumor among adolescent patients and typically occurs near the joint [17], and tumor endoprostheses are usually used to reconstruct the bone defect created by tumor resection. Although joint arthroplasty can fill large defects, when artificial joint replacement is performed on younger patients, there is a high risk for complications such as loosening or implant failure due to its long-term postoperative course, potentially requiring an invasive revision procedure [18].

Autologous bone grafting is a method of using one’s own bone to fill a defect. Since the graft uses bone obtained from the same patient receiving the graft, the graft material can be harvested relatively safely from a site with low donor-site morbidity, even if there is a certain amount of bone defect that occurs at the donor site. The iliac crest and fibula are often used as donor sites. There are no biocompatibility issues, as the graft uses bone from the patient’s own body. However, there is a finite supply of bone that can be harvested from the patient, and the donor site is damaged in order to harvest the bone for grafting, which may cause problems such as bleeding, infection, and pain at the donor site [19,20,21,22].

Allogeneic bone grafting is a treatment method wherein the bones of a donor are heat-treated, cryopreserved, and filled in the bone defect. The bones used are obtained from cadavers and surplus bone tissues that are no longer needed in surgery such as artificial joint replacement. Although there is an ample supply of bone in countries with well-developed bone bank services for the preservation and utilization of cadaver bones [23,24,25,26,27], in countries like Japan where human remains after cremation are buried, cultural and religious practices often prevent the collection of donor bone from cadavers. Thus, the donation of allogeneic bone can be in short supply at bone banks in these countries [28]. The resorptive and regenerative properties of allogeneic bone is not as effective as that of autologous bone; however, it not only serves as a scaffold, but also contains growth factors such as bone morphogenetic protein (BMP) that remain after heat treatment and enable osteoinduction [29,30,31]. The commercialization of the human demineralized bone matrix (DBM) has developed in recent years, and scaffolds that retain the ability to induce bone using growth factors have become widely available [32,33,34,35].

Bone regeneration with synthetic bone is performed using synthetic bone made of calcium phosphate bioceramics such as β-tricalcium phosphate (βTCP), carbonate apatite, and hydroxyapatite. These synthetic bones form a porous body and act as a scaffold for bone regeneration. Although the synthetic bone itself functions as a scaffold, it is incapable of osteoinduction and is unsuitable for the treatment of large bone defects.

The induced membrane technique can be used to treat relatively large bone defects by mixing synthetic bone with autologous bone [15,36]. However, because a membrane must be induced for bone regeneration, a separate operation is required prior to the grafting of synthetic and autologous bone. Since autologous bone is used, there is a limit to the number of defects that can achieve regeneration.

As noted above, there are numerous treatment options for bone defects. Each treatment option presents advantages and disadvantages, and there is still room for development and research. In particular, treatment with synthetic bone is less invasive to patients, and improvement in efficiency and expansion of indications for bone regeneration using synthetic bone are strongly desired in bone-regeneration treatment. In this review, we will introduce the current clinical use of artificial bones composed of calcium phosphate bioceramics in addition to the recent research and development of artificial bones composed of calcium phosphate bioceramics, biodegradable polymers, and their composites.

## 2. Clinical Application and New Basic Research on Synthetic Bone Composed of Calcium Phosphate Bioceramics

Currently available synthetic bones used in clinical practice are mainly made of calcium phosphate such as hydroxyapatite and βTCP. Granular and block-type synthetic bones are porous [37] (Figure 1a). Osteoblasts and osteoclasts invade the pores and proliferate, promote the resorption of the synthetic bone and subsequent osteogenesis, and allow a gradual replacement with the patient’s own bone. The bone defect caused by a bone tumor is initially filled with synthetic bone, and the regeneration of autologous bone can ultimately be expected. The treatment is mainly used for cancellous bone defects when the cortical bone remains intact. Autologous bone may be used in combination for large defects and bone defects near the articular surface.

Figure 2a is a plain radiography image of a patient with a large aneurysmal bone cyst (ABC) in the tibia. Since the bone defect is close to the articular surface, a layer of autologous iliac cortical bone graft is placed under the subchondral bone in the deep layer of the articular surface after curettage of the tumor, and block-shaped and granular βTCP synthetic bones are filled underneath the autologous graft. The defect is fixated using a titanium alloy plate and screws for reinforcement (Figure 2b,c). Five years after the operation, the autologous iliac bone graft demonstrates osseous integration with the subchondral bone, and the syn-thetic bone is replaced with the autologous bone; however, residual synthetic bone is still visible in the center (Figure 2d).

The HA/collagen sponge composite is marketed as a synthetic bone for the treatment of bone defects. The composite combines both the osteoconductivity of HA and rapid degradation of collagen to enable a quick replacement with autologous bone [11,38]. However, the initial mechanical strength of material is weak when used as a synthetic bone, and inflammatory reactions including exudation, redness of surgical wounds, and swelling may potentially occur in bones near the superficial layer such as finger bones [11]. Therefore, the implantation site is limited, and careful monitoring is required for postoperative recovery. When the scaffold is infiltrated with blood or tissue fluid in the body, the scaffold becomes soft enough to fill the void tightly according to the shape of the bone defect (Figure 1a).

Figure 3 shows a patient with ABC of the humerus that recurred after two surgeries (curettage, βTCP synthetic bone grafting). For the third surgery, synthetic bone made of a HA/collagen composite was filled after curettage (Figure 3a–e). Five years after the operation, no tumor recurrence was observed, and regeneration of the autologous bone was observed; however, a growth disorder of the humerus occurred due to the effects of surgery during childhood, and the affected humerus became shorter compared to the unaffected side (Figure 3g,h). Although synthetic bone made of a HA/collagen composite provides good intraoperative handling, its high radiolucency is a disadvantage that makes postoperative radiologic confirmation difficult with plain radiography.

Carbonate apatite is an inorganic component of bone tissue. Unlike hydroxyapatite, it is not sintered at high temperatures; thus, it has low crystallinity and is quickly absorbed and replaced in the biological environment with bone tissue. In animal experiments, Fujisawa et al. [39] and Mano et al. [40] showed that artificial bone composed of carbonate apatite replaces autologous bone more efficiently than artificial bone composed of hydroxyapatite. There have been advancements in the clinical application of artificial bones composed of carbonate apatite, especially in the dental field. As an example, artificial bones made of granular carbonate apatite are currently used for periodontal regenerative therapy [41,42].

In the treatment of bone defects with synthetic bones, the presence of residual cortical bone is desirable. However, if the induced membrane technique is used, even a circumferential defect can be regenerated. The surgical procedure of the induced membrane technique is carried out in two steps. In the first step, the bone defect is filled with bone cement that acts as a spacer to close the wound. A periosteal-like tissue is subsequently formed around the spacer. A second surgery is performed 1–2 months later, in which the spacer is removed, and a mixture of autologous and granular synthetic bone are mixed and filled into the bone defect in its place [15]. The treatment of large bone defects is made possible by forming an induced membrane and mixing autologous and synthetic bone. Although we performed extracorporeal irradiation to kill malignant tumor cells in the femur and re-implanted the bone for cases with malignant femoral tumors [43], the irradiated bone showed a poor bony union rate, and bony union was sometimes not achieved. Following bone reconstruction with the extracorporeal irradiation and the re-implantation technique for patients who exhibited nonunion or intramedullary nail failure, the femur was regenerated using the induced membrane technique. The spacer was removed during the second operation, filled with synthetic and iliac bone graft, and new bone was regenerated in the femoral bone defect and fixated with an intramedullary nail and an osteosynthesis plate. In addition, good bone formation was achieved with the induced membrane technique even in cases with a large bone defect that occurred due to a benign bone tumor of the tibia and extensive damage to the cortical bone that resulted from tumor curettage (Figure 4).

Synthetic bone made of calcium phosphate, which has already been used in clinical applications, is a convenient and powerful scaffold for bone-regeneration therapy. However, there is still room for improvement, as there are limitations in terms of the site and size of bone defects that are suitable for treatment, a potential need for the combined use of autologous bone grafts, a possible need for multiple surgeries, and a delay in weight-bearing ambulation for cases of the lower extremities.

In recent years, considerable attention has been focused on octacalcium phosphate (OCP) as a new calcium phosphate material for artificial bone. OCP is believed to be a precursor of hydroxyapatite in bone tissue and is a substance observed during the growth stage of hydroxyapatite crystals [44]. OCP encourages bone formation by promoting the differentiation of osteoblasts while also inducing the formation of osteoclasts, which allows the material to be rapidly absorbed. OCP provides better osteoconductivity than conventional artificial bone made of calcium phosphate [45]. However, OCP alone has poor shaping and handling properties; therefore, a body of research has weighed in on the development of composites with other materials.

## 3. Research and Development of Biodegradable Polymers

In regenerative medicine, biodegradable polymers are used as scaffolds that are gradually replaced with the patient’s own tissues, in addition to their roles as carriers of growth factors and cells. Langer and Vacanti succeeded in seeding bovine articular chondrocytes onto a scaffold composed of polyglycolic acid-polylactic acid in the shape of a human auricle and transplanting it subcutaneously into the dorsum of an immunocompromised mouse [46,47]. The scaffold is envisaged to play a role in regenerative medicine that is as important as pluripotent cells that can be engrafted in tissues [48]. Factors that influence the performance of a scaffold in regenerative medicine include its material (bio/tissue affinity, chemical properties), composite, three-dimensional structure, added cells, and signaling molecules. The following sections will describe each factor.

### 3.1. Material of the Polymer

Bio-derivative polymers exhibit exceptional biosafety and are being studied as potential biomaterials [49]. Collagen is abundantly present in the living body as a component of bone and in soft tissues including bone, cartilage, ligaments, and tendons. In bone, more than 90% of the matrix proteins are composed of collagen. When bone is formed, calcium phosphate is deposited on osteoblast-produced collagen fibers to create a tough bone matrix. [50]. Since collagen natively plays an important role as an extracellular matrix for hard and fibrous tissues of the human body, extensive research has been conducted for its use as a scaffold in regenerative medicine, including evaluative studies for the myocardium [51], bladder [52], and ligament [53]. Bone-tissue regeneration is being studied clinically in animal models. Carstens et al. [54] described the use of collagen as a scaffold to regenerate non-weight-bearing bones, including the maxilla and mandible.

Gelatin is a triple-helix structure derived from denatured and decomposed polypeptide chains of collagen fibers. Although the most common applications of gelatin are for culinary and cosmetic usage, the material is also used in the medical field such as capsules for pharmaceuticals, as well as as a hemostatic agent [55] and an embolic substance for arterial embolization [56]. Gelatin is also being studied as a scaffold in regenerative medicine. Yokota et al. [57] created a scaffold from a gelatin sponge coated in poly(D,L-lactic-co-glycolic acid) as a carrier for recombinant human bone morphogenetic protein (rhBMP)-2 and implanted the scaffold in the dorsal subcutaneous tissue of a rat to induce ectopic bone formation. Rohanizadeh et al. [58] reported the possible use of a gelatin sponge, a commercially available hemostatic agent, as a scaffold. They cultured human MG-63 osteoblast-like cells on a gelatin sponge and observed their cell number, alkaline phosphatase (ALP) activity, and cell invasion into the pores of the sponge.

Although collagen is a gold standard for scaffolds made of animal-based polymers in regenerative medicine, various other materials have been considered for use as a scaffold, such as cellulose obtained from plant polysaccharides [59], chitosan derived from the exoskeleton of crustaceans such as shrimp and crab [60,61], and hyaluronic acid [62,63], which is also a component of articular cartilage and articular fluid.

Natural polymers are used in foods and cosmetics and are considered biomaterials with high biocompatibility. In basic research, many reports describe the feasibility of using natural polymers as a scaffold in bone regenerative medicine due to their high biocompatibility and rapid degradability. However, there are concerns about risks such as immune reactions due to disease transfer and xenogenicity [64,65]. Clinical applications of natural polymers should warrant caution, as there have been reports of allergies due to injections and foods, inflammation, and pulmonary complications [66,67,68].

A biodegradable synthetic polymer is capable of being hydrolyzed and absorbed in vivo. The use of synthetic polymers as scaffolds for bone regeneration is being investigated. We developed and evaluated the performance of a scaffold using polylactic acid-p-dioxanone-polyethylene glycol block copolymer (PLA-DX-PEG), which is a biodegradable polymer, as a carrier of rhBMP-2 [69] (Figure 5a). We succeeded in inducing the formation of ectopic bone under the dorsal fascia of mice using this scaffold (Figure 5a–d). This scaffold was also able to repair a critical-sized bone defect in the rabbit ulna [48] (Figure 5e). Other common synthetic polymers that have been evaluated as materials for scaffolds include poly-L-lactic acid (PLA), polycaprolactone (PCL), polylactic-co-glycolic acid (PLGA), and poly (vinyl alcohol) (PVA).

Although the PLA-DX-PEG we previously studied is a copolymer of PLA and PEG, PLA itself is also being evaluated as a scaffold. PLA is a biodegradable polymer widely used in food trays and agricultural films. Zhang et al. [70] created a collagen/PLA scaffold in which collagen was combined with the layer of nanofiber of PLA. From the bone marrow-derived mesenchymal stem cells (BMSCs) cultured onto this scaffold, a gene expression of osteocalcin (OCN), which is a bone formation marker, was observed stronger than that of the BMSCs cultured onto collagen scaffolds. They also filled the osteochondral defects created in the femur of rabbits with the collagen/PLA scaffold, and an evaluation of the scaffold using the Visual Histological Assessment Scale of the International Cartilage Repair Society [71] demonstrated better regeneration of the subchondral bone than the group filled with a scaffold made of collagen alone.

PCL is a thermoplastic with a low melting point and a polymer of ε-polycaprolactone [72] with exceptional biocompatibility [73]. Wang et al. [74] added nanosilicates to PCL to prepare scaffolds comprising nanofibers and cultured MC3T3-E1 cells, which are osteoblast cell lines. On this scaffold, the cell viability and ALP activity of MC3T3-E1 cells increased according to the amount of nanosilicates.

Yang et al. [75] created and evaluated a scaffold consisting of nanofibers in which nanosilicates were combined with PLGA. They cultured osteoblast-like cells (SaOS-2 cells) on this scaffold and assessed their differentiation into bone using Alizarin Red S staining and ALP activity. As a result, it was shown that nanosilicate/PLGA scaffold promoted the better differentiation of SaOS-2 cells into bone compared to a scaffold made of PLGA alone.

PVA is highly hydrophilic and easily dissolves in vivo [76]. Kim et al. conducted an experiment in which MG-63 osteoblast-like cells were cultured on implants formed using 3D printing using a composite of gelatin and PVA [77]. They showed that ALP activity and calcium deposition with MG-63 cells on gelatin/PVA scaffolds were highest when the weight ratio of gelatin to PVA was 1:1.

Biodegradable polymers in bone tissue scaffolds should ideally be completely replaced by autologous bone by being dissolved and absorbed in vivo. Rohanizadeh et al. proposed a gelatin sponge scaffold with a PLGA coating that reduced its degradation rate and proliferation of MG-63 osteoblast-like cells on the implant; however, the decreased degradation rate was not due to the material alone. Changes in the shape of the implant surface and the ease with which MG-63 osteoblast-like cells adhered to the PLGA coated on the surface were also suggested to affect degradation [58]. Hsieh et al. [78] created a scaffold that combined PVA with curdlan, which is also used as a food additive, and evaluated its degradability with the degrading enzymes lipase and lysozyme. The results showed that the degradation rate of the PVA scaffold did not have a dose-dependent effect on the amount of curdlan added, and the results differed depending on the type of enzyme. The balance between the rate of degradation and bone formation is important in scaffolds. The optimal rate of degradation for each material needs to be demonstrated with in vivo testing under varying conditions.

A summary of the literature discussed in this section is shown in Table 1.

### 3.2. Biodegradable Polymer and Calcium Phosphate Bioceramics Composites

To take advantage of both the bioabsorbability of biodegradable polymers and the osteoconductivity of calcium phosphate bioceramics, composites of these materials have been created and experimentally evaluated.

Venugopal et al. [79] created a composite of Type I collagen and HA, on which human fetal osteoblast cells were cultured. In Alizarin Red S staining for assessing calcification [80], cultures on this HA–collagen composite scaffold showed more vivid staining compared to the HA-free collagen fiber scaffold [79]. Yeo et al. [81] created a three-dimensional porous composite of βTCP and polycaprolactone (PCL), which was filled with collagen nanofibers to form scaffolds. The MTT (3- (4,5-dimethylthiazol-2-yl) -2, 5-diphenyl tetrazolium bromide) assay [82], used to assess the cell proliferation of the human MG-63 osteoblast-like cell, demonstrated a higher cell proliferation in this scaffold compared to the βTCP/PCL composite scaffold without collagen nanofibers. Kane et al. [83] evaluated the effects of HA concentration, HA shape (powder-like or fibrous), and scaffold porosity in HA/collagen scaffolds. They reported that the compressive modulus of the scaffold increased as the vol% of HA increased for scaffolds with 85% porosity. Regarding the shape of the HA particle, when the HA content was 20 to 60 vol%, the compressive modulus was higher in the fibrous HA particle, but at 80 vol%, the compressive modulus was higher in the powder-like HA particle. When the porosity of the scaffold was set to 90%, the compressive modulus of both powder-like and fibrous HA particles decreased significantly.

Enayati et al. [84] developed a PVA/HA scaffold that combines PVA with nanoparticles of HA. They cultured MG63 cells on a PVA/HA scaffold and evaluated their effect on bone formation using Alizarin Red S staining and ALP activity. As a result, the PVA/HA scaffold promoted better differentiation of MG63 cells into osteoblasts than the scaffold with scaffold with alone. Although PVA is a substance with exceptional biocompatibility, PVA itself is bio-inert [85], and it is speculated that the additional effect of HA nanoparticles promoted differentiation into osteoblasts.

Hamai et al. created a composite of gelatin and OCP to produce gelatin/OCP granules, which were subsequently hardened with gelatin on a disc to create an implant. We conducted an experiment to repair critical-sized calvarial defects in rats using an implant of the same morphology with OCP granules solidified without gelatin and that with gelatin as a control [86]. As a result, implants made from gelatin/OCP granules showed more active new bone formation in the rat calvarial defects, higher orientation of apatite, and higher quality bone regeneration. They suggested that gelatin/OCP promotes hydrolysis and is involved in the improvement of osteogenic properties.

Ruckh et al. created a HA/PCL scaffold and evaluated bone formation markers in rat marrow stromal cells [87]. They created scaffolds containing 1wt% and 10wt% of HA, respectively, and compared their osteogenic properties with a PCL scaffold that did not contain HA. The ALP activity increased dose-dependently with HA content, but the gene expression of Type I collagen and osteopontin (OPN) decreased as HA content increased. Changes in the HA content of composites affected the response of bone formation markers; however, it is necessary to evaluate each type of scaffold to determine what effect it has on actual bone formation. Various other composites have been developed in recent years, including collagen/β-TCP [88], collagen/OCP [89], PLA/HA [90], PLA/β-TCP [91], PLA/OCP [92], and PLA/PCL/HA [93].

Combining biodegradable polymers with calcium phosphate bioceramics enables the creation of scaffolds that take advantage of the strengths of each material. Many combinations of scaffolds have been studied, and the results showed promising bone regeneration that was superior to that of single-material scaffolds. For polymer composites, the type of material, compositing method, and three-dimensional structure are factors related to bone regeneration. However, there are numerous combinations, and the ideal makeup and conditions of composites for bone formation remain unclear.

A summary of the literature discussed in this section is shown in Table 2.

### 3.3. Three-Dimensional Structure of Synthetic Bonessc

For bone-regeneration scaffolds, not only the material but also the three-dimensional structure is an important factor, and various forms of scaffolds are being investigated.

Like the PLA-DX-PEG scaffold we used, there are other hydrogel scaffolds. We were able to combine rhBMP-2 with a PLA-DX-PEG scaffold to form ectopic bone on the dorsum of mice and regenerate bone in critical-sized bone defects of rabbit ulna [48]. Other scaffolds reported in the literature include gelatin hydrogel [94] and gelatin and chitosan composite hydrogel [95].

A sponge-like scaffold can be made by freeze-drying the hydrogel material (Figure 6a). Takeda et al. [96] created a sponge-like implant from a collagen/rhBMP-2 composite and reconstructed rat collagenellae (ossicles). Takahashi et al. [97] cultivated rat mesenchymal stem cells (MSCs) on a gelatin/β-TCP composite sponge and demonstrated using scanning electron microscopy (SEM) that MSCs invaded and adhered to the pores of the sponge (Figure 6b). By making the scaffold a porous body, cells are more likely to enter, and a superior tissue regeneration is observed compared to those in the form of a dense, compacted body.

Fiber-shaped implants can also serve as good scaffolds by controlling their three-dimensional structures [98]. Electrospinning is a method of fiber production that charges and ejects a polymer solution through a nozzle under a high-voltage electric field to produce nano-sized fibers [99]. Lee et al. [100] created a collagen fiber and PCL composite with a diameter of approximately 350 nm. By hardening collagen nanofiber with exceptional cell adhesion and proliferation using PCL, it was possible to increase the mechanical strength of the scaffold. They cultured MG63 cells on this scaffold and demonstrated better cell proliferation using MTT assay compared to a scaffold made of PCL alone. Both the collagen/PLA composite scaffold described by Zhang et al. [70] and the PVA/HA composite scaffold described by Enayati et al. [84] are fibrous scaffolds produced through electrospinning (Figure 6c).

By creating a scaffold using 3D printing, it has become possible to control its fine structure. The PLGA/HA/chitosan scaffolds described by Deng et al. [101] were made using 3D printing, and their pore size was approximately 430 μm. Zhang et al. [102] made a PTG implant in which graphene oxide (GO) was combined with PLGA and βTCP using 3D-printing technology. This scaffold was a lattice-structure implant with a pore size of 400 ± 50 μm (Figure 6d,e). MSCs of rats were cultured onto this scaffold, and increased gene expression of bone-formation markers ALP, OCN, and osteopontin (OPN) were observed. They also used this scaffold to repair critical-sized cranial defects in rats [102].

A summary of the literature discussed in this section is shown in Table 3.
Figure 6Three-dimensional structure of biodegradable polymer. (**a**) Scanning electron microscopy (SEM) image of a scaffold prepared through freeze-drying Type I collagen; (**b**) SEM image of a gelatin/β-TCP composite scaffold. Image is modified from a study by Takahashi et al. Reproduced with permission from Elsevier, 2005 [97]; (**c**) SEM image of a PVA/HA composite scaffold created using the electrospinning method. Image is modified from a study by Enayati et al. Reproduced with permission from John Wiley and Sons, 2018 [85]; (**d**) photograph of a PLGA/βTCP/GO (PTG) composite scaffold created using 3D printing; (**e**) SEM image of a PTG composite scaffold. Images are modified from a study by Zhang et al. [102].
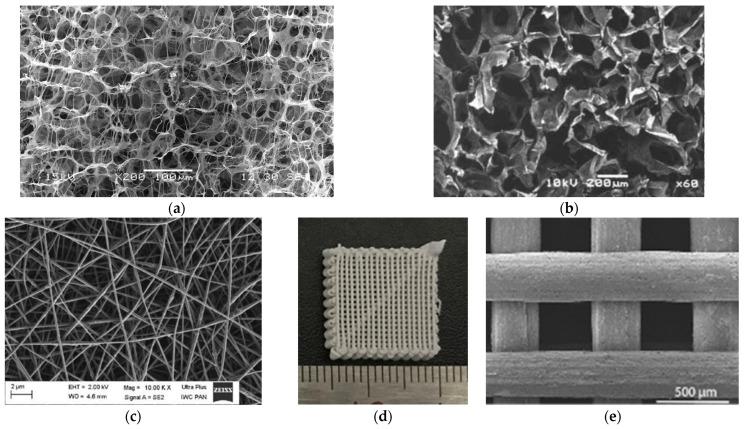


### 3.4. Cells and Signaling Molecules

Calcium phosphate bioceramic scaffolds (TCP, HA) are osteoconductive and are currently used clinically in the treatment of bone defects. Relatively small bone defects can be repaired by simply filling voids with this scaffold. However, there is a limit to the size of bone defects that can be treated with this scaffold alone.

Unlike calcium phosphate-based scaffolds that are osteoconductive, biodegradable polymers are not osteoconductive. Therefore, it is necessary to improve the efficiency of bone regeneration by adding signaling molecules and cells to these scaffolds in order to increase their osteoconductivity.

Among the signaling molecules that promote bone formation, BMP-2 is the most commonly used. BMP-2 induces bone formation and has been put to practical use in the treatment of fractures and bone defects. In order for BMP-2 to act efficiently in fractures and bone defects, it is important for scaffolds to act as a carrier and drug delivery system (DDS) for BMP-2. In the in vitro test of the PLGA/HA/chitosan scaffold reported by Deng et al. described in the previous section, when BMP-2 is added to PLGA/HA/chitosan scaffold and immersed in a culture medium, the scaffold takes more than 2 weeks to disintegrate and to gradually release the BMP-2 over a month period or more. In an in vivo study, Deng et al. [101] reported that this scaffold successfully repaired a bone defect created in a rabbit mandible.

Other signaling molecules such as BMP-6 and BMP-7, which are part of the BMP family, and vascular endothelial growth factor (VEGF), which is an angiogenesis factor/tissue growth factor, are expected to be used in clinical applications [103,104].

In the body, when scaffolds are placed on the affected area, bone-forming cells from surrounding tissues enter the scaffold and proliferate to form bone. However, to repair a large bone defect, it can be expected that replacement with new bone will be accelerated by engrafting cells in a scaffold in advance. The cells combined with scaffolds are expected to produce the aforementioned signaling molecules and to differentiate into target organs/tissues to be regenerated. In recent years, many studies have been conducted using iPS cells. iPS cells can be produced by introducing several types of genes called Yamanaka factors into somatic cells collected from the skin and demonstrate pluripotency that allows them to differentiate into any cell. Since they can be differentiated into various cells, they are expected to be applied clinically in regenerative medicine, treatment of intractable diseases, and cancer treatment. In bone regeneration, iPS cells are also being investigated as cells to be added to scaffolds.

The cells seeded on scaffolds for bone regeneration do not need to have pluripotency like iPS cells, and it is sufficient if they differentiate into bone tissue. Therefore, considerable research on BMSC, which is a progenitor cell of osteoblasts, has been conducted [105,106]. BMSC from a patient can be relatively easily obtained in abundance from the iliac bone marrow of the patient with minimal invasiveness. BMSC can be differentiated into osteoblast progenitor cells by culturing in a bone-forming medium containing β-glycerophosphate and dexamethasone. Studies are being conducted to promote bone formation by culturing BMSC on scaffolds and differentiating it into osteoblast progenitor cells. One research report has recently described the addition of BMSC to a scaffold made of chitosan/HA and PCL/PLA composites [107].

A therapeutic method of adding platelet-rich plasma (PRP) to scaffolds is also being investigated. PRP can be easily obtained by centrifuging the patient’s peripheral blood. PRP contains growth factors such as VEGF, insulin-like growth factor (IGF), platelet-derived growth factor (PDGF), and transforming growth factor beta (TGF-β). These growth factors are not expected to act as cells that differentiate into tissues but are rather expected to be used as a DDS for patient-derived signaling molecules [61,108]. Cheng et al. [109] treated critical-sized cranial bone defects in rats with a silk fibroin/PCL composite scaffold augmented with PRP.

A summary of the literature discussed in this section is shown in Table 4.

## 4. Conclusions

The performance of biodegradable scaffold in bone regeneration is associated with its material, three-dimensional structure, and added cells and signaling molecules; however, there are innumerable combinations that can yield varying results (Figure 7). Synthetic bone made of calcium phosphate bioceramics has already been used clinically in human patients. Although synthetic bone made of calcium phosphate bioceramics is a convenient therapeutic device, there are some disadvantages such as the time it takes to be replaced with bone, the inability to perform early loading in weight-bearing bone, limitation to the size of bone defects that can be treated, and the need to harvest autologous bone when the defect is large.

Various research endeavors on synthetic bones made of biodegradable polymers have been conducted, but their performance is still inferior to that of calcium phosphate bioceramic material, which has osteoconductivity. Synthetic bone made of a HA/collagen composite—a composite of collagen, a biodegradable polymer, and HA, a calcium phosphate bioceramic—is one successful example. By clarifying the optimum combination of material properties, structural characteristics, and cells and signaling molecules, the development of an ideal scaffold with high bone-regeneration efficiency can be expected in future studies.

## Figures and Tables

**Figure 1 bioengineering-11-00180-f001:**
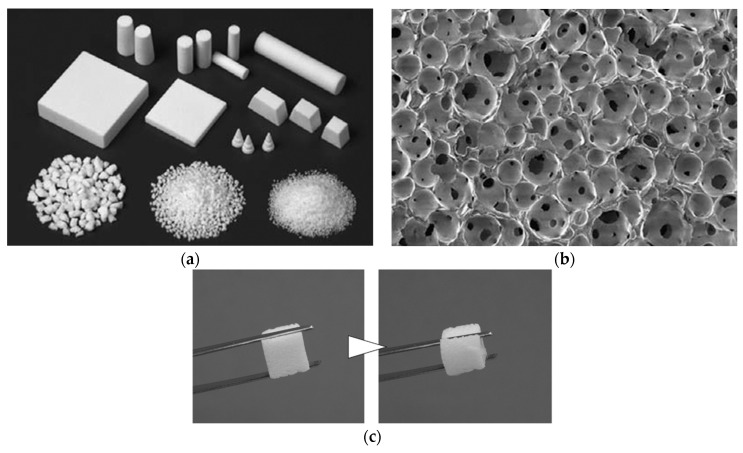
Synthetic bone made of hydroxyapatite (HA). (**a**) Photograph of synthetic bone made of HA. There are block-shaped, cylindrical, and granular products, which are used according to the size and shape of the bone defect. (**b**) Scanning electron microscopy (SEM) image of synthetic bone made of HA. The synthetic bone forms a porous body with micrometer-sized pores. Images are modified from a study by Yoshikawa et al. Reproduced with permission from The Royal Society, 2009 [37]. (**c**) Photograph of synthetic bone made of a HA/collagen composite (left). The synthetic bone becomes soft when it contains water. Images are modified from a study by Sotome et al. [11].

**Figure 2 bioengineering-11-00180-f002:**
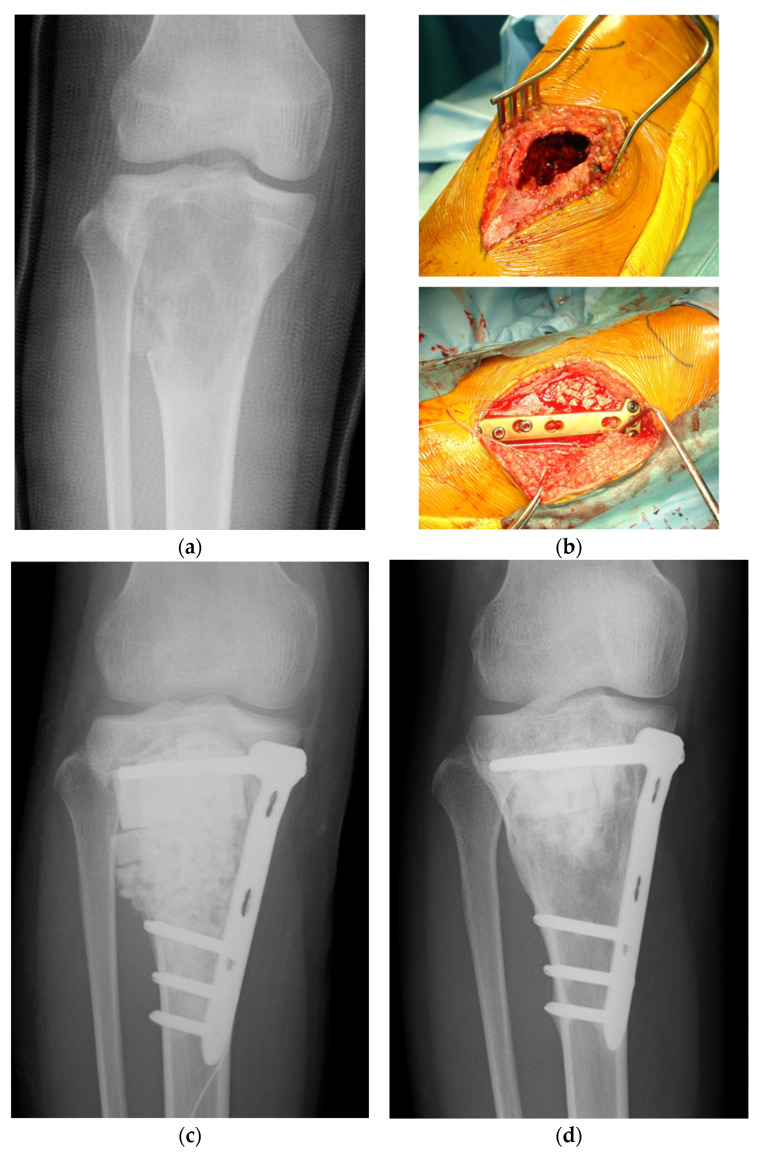
A case with a tibial aneurysmal bone cyst (ABC). (**a**) Preoperative plain radiography image. Radiotransparency is observed from the epiphysis to the epiphysis of the tibia. There is a pathological fracture at the tumor site, and the fixation is performed with a cast. (**b**) Intraoperative photograph following tumor curettage. After curettage of the tumor, autologous iliac bone is grafted just below the articular surface, and most of the remaining space is filled with synthetic bone made of βTCP and reinforced with plates and screws. (**c**) Plain radiography immediately after surgery. It is observed that the bone defect where the tumor curettage was performed is filled with block-shaped and granular synthetic bones. (**d**) Plain radiography at 5 years after surgery. The autologous iliac graft just below the articular surface which shows bony union. The synthetic bone has been replaced with autologous bone, but some of the large block-shaped synthetic bone remains. No tumor recurrence is observed.

**Figure 3 bioengineering-11-00180-f003:**
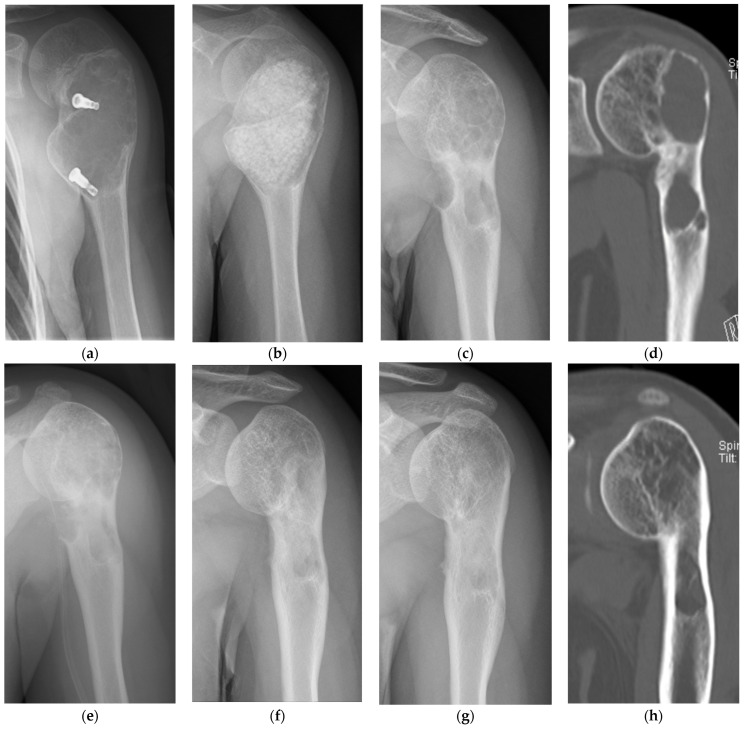
A case of ABC of the humerus. (**a**) A diagnosis of a simple bone cyst was made by a previous doctor, and decompression was performed with a cannulated screw made of hydroxyapatite, but the tumor did not shrink and a radiotransparency was observed. (**b**) The tumor was curattaged at our hospital and filled with granular βTCP synthetic bone. (**c**) Plain radiography at 1 year after synthetic bone filled with βTCP. The synthetic bone has been resorbed and the tumor has recurred. (**d**) CT image at the time of tumor recurrence. Two recurrent tumors are found at the metaphysis and at the diaphysis. (**e**) Plain radiography after reoperation at our hospital. After tumor curettage, the synthetic bone made of collagen/HA composite was filled. The synthetic bone made of the collagen-HA composite has high radiotransparency and is less visible compared to the βTCP synthetic bone. (**f**) Plain radiography at 1 year after reoperation. The autologous bone is regenerating. (**g**) Plain radiography at 5 years after reoperation. No recurrence is observed. (**h**) CT image at 5 years after reoperation. A shadow of regenerated cancellous bone is observed at the site where there was a bone defect due to the tumor. Deformity remains at the metaphyseal end of the humerus.

**Figure 4 bioengineering-11-00180-f004:**
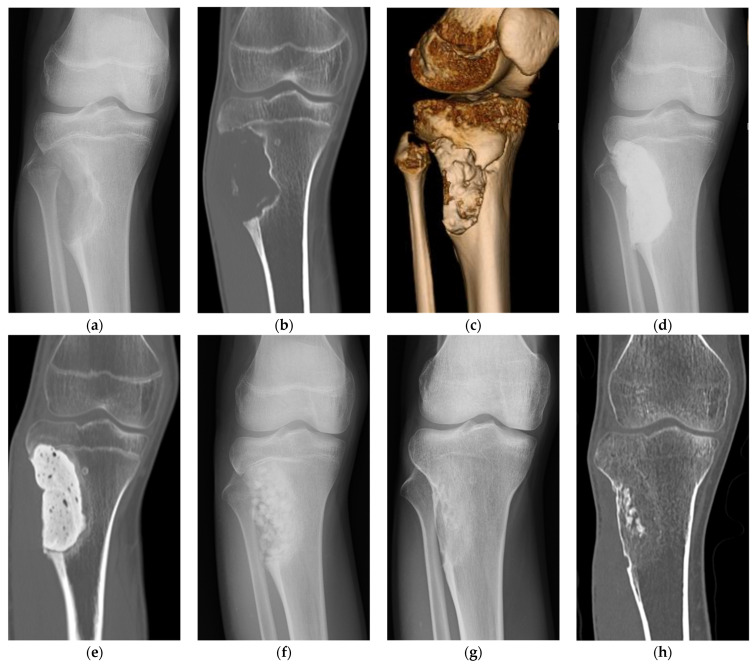
A case with chondromyxoid fibroma. (**a**) Preoperative plain radiography. A bone defect has occurred at the tumor site proximal to the lateral tibia. (**b**) Preoperative CT image. Due to the tumor, bone defects including the outer cortical bone are observed. (**c**) Preoperative 3D CT image. Bone defects at the tumor site are observed as depressions. (**d**) Simple X-ray image immediately after the 1st stage operation with induced membrane technique. The bone defect is filled with bone cement. (**e**) CT image immediately after the 1st stage operation. (**f**) Six weeks after the first surgery, the second surgery is performed. Plain radiography after second surgery. Bone cement is removed and filled with a mixture of granular βTCP synthetic bone and autologous iliac cancellous bone. (**g**) Plain radiography at 3 years after surgery. Resorption of synthetic bone is progressing, and autologous bone is being regenerated. (**h**) CT image at 3 years after surgery. Autologous bone, including the cortical bone, is regenerating. Some synthetic bone remains inside.

**Figure 5 bioengineering-11-00180-f005:**
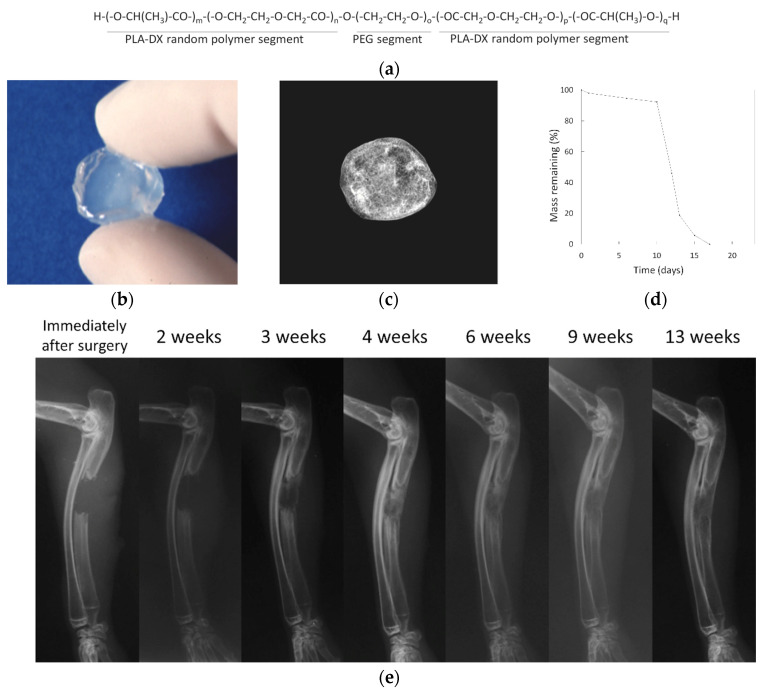
(**a**) Structural formula of polylactic acid-p-dioxanone-polyethylene glycol block copolymer (PLA-DX-PEG). (**b**) Photograph of PLA-DX-PEG hydrogel. (**c**) Ectopic bone formed under the dorsal fascia of mice by adding rhBMP-2 to the PLA-DX-PEG scaffold. (**d**) Dissolution curve of PLA-DX-PEG polymer. The polymer was immersed in PBS at 37 °C and weighed. The weight was gradually reduced over 10 days and completely dissolved within 20 days. Images are modified from a study by Saito et al. [69]. (**e**) rhBMP-2 was added to the PLA-DX-PEG scaffold to treat a critical-sized bone defect in the rabbit ulna. Approximately 3 months after the operation, the bone defect was repaired. Images are modified from a study by Aoki et al. [48].

**Figure 7 bioengineering-11-00180-f007:**
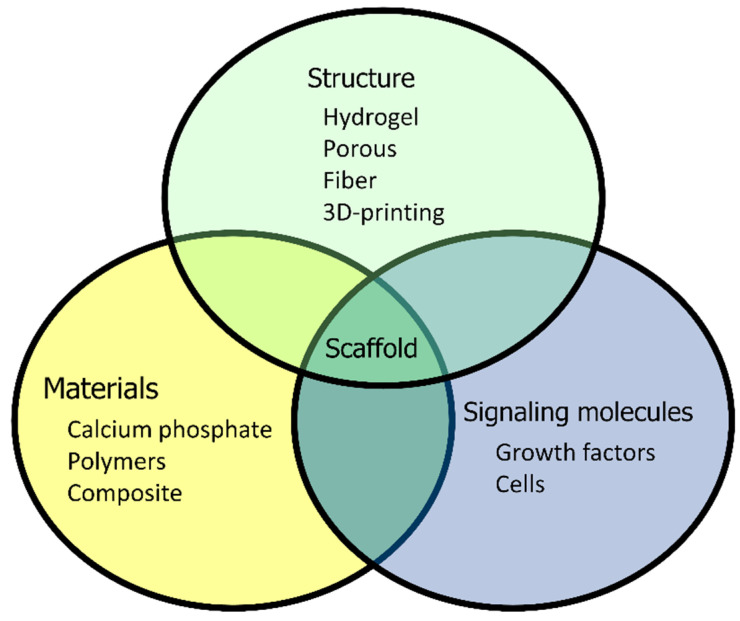
Factors that affect the performance of biodegradable scaffolds.

**Table 1 bioengineering-11-00180-t001:** Representative biodegradable polymers.

Author, Year	Natural or Synthetic	Polymer	Characteristics
Castens et al., 2005 [54]	Natural	Collagen	Used to repair porcine mandibular bone defect
Yokota et al., 2001 [57]	Natural	Gelatin	Coated in poly(D,L-lactic-co-glycolic acid)
Rohanizadeh et al., 2008 [58]	Natural	Gelatin	Used to culture human osteoblast-like cells
Chakraborty et al., 2001 [59]	Natural	Cellulose	Non-woven nanofibrous scaffolds made by electrospinnig
Sharifi et al., 2018 [60]	Natural	Chitosan	Composite with PCL
Liu et al., 2018 [61]	Natural	Chitosan	Composite with HA
Yan et al., 2018 [62]	Natural	Hyaluronic acid	Used as carrier for BMP-2 to form ectopic bone in rat
Paidikondala et al., 2019 [63]	Natural	Hyaluronic acid	Composite with hydrazone
Saito et al., 2001 [69]	Synthetic	PLA-DX-PEG	Used to form ectopic bone in dorsum of mouse
Aoki et al., 2020 [48]	Synthetic	PLA-DX-PEG	Used to repair ulnar segmental bone defect of rabbit
Zhang et al., 2013 [70]	Synthetic	PLA	Composite with collagen
Wang et al., 2018 [74]	Synthetic	PCL	Composite with nanosilicate
Yang et al., 2018 [75]	Synthetic	PLGA	Composite with nanosilicate
Kim et al., 2018 [77]	Synthetic	PVA	3D-printed scaffold
Hsieh et al., 2018 [78]	Synthetic	PVA	Used to evaluate biodegradation of 3D scaffolds

PCL: polycaprolactone, HA: hydroxyapatite, BMP: bone morphogenetic protein, PLA-DX-PEG: poly lactic acid-p-dioxanone-polyethylene glycol block copolymer, PLA: poly-L-lactic acid, PLGA: polylactic-co-glycolic acid, PVA: poly (vinyl alcohol).

**Table 2 bioengineering-11-00180-t002:** Composite of biodegradable polymer and calcium phosphate bioceramics.

Author, Year	Biodegradable Polymer	Calcium Phosphate	Characteristics
Venugopal et al., 2008 [79]	Collagen	HA	Used to evaluate calcification caused by human fetal osteoblast cells
Yeo et al., 2011 [81]	PCL, collagen	βTCP	Used to culture human osteoblast-like cells
Kane et al., 2015 [83]	Collagen	HA	Used to evaluate the compressive modulus of the scaffold
Enayati et al., 2018 [84]	PVA	HA	Used to culture human osteoblast-like cells
Hamai et al., 2022 [86]	Gelatin	OCP	Used to repair critical-sized calvarial defect of rat
Ruckh et al., 2012 [87]	PCL	HA	Used to evaluate osteogenic potential according to HA content
Mohseni et al., 2018 [88]	Collagen	βTCP	Used to repair ulnar segmental bone defect of rabbit
Suzuki et al., 2020 [89]	Collagen	OCP	Used to compare HA and βTCP
Li et al., 2023 [90]	PLA	HA	Used to evaluate osteogenic potential of rat BMSCs
Zarei et al., 2024 [92]	PLA	OCP	Composite with Ti6Al4V to evaluate compressive strength
Hassanajili et al., 2019 [93]	PLA, PCL	HA	Used to evaluate porosity and compressive modulus with blending ratio of each material

HA: hydroxyapatite, PCL: polycaprolactone, TCP: tricalcium phosphate, PVA: poly (vinyl alcohol), OCP: octacalcium phosphate, PLA: poly-L-lactic acid.

**Table 3 bioengineering-11-00180-t003:** Three-dimensional structure of scaffolds.

Author, Year	Materials	Structure
Saito et al., 2001 [69]	PLA-DX-PEG	Hydrogel
Aoki et al., 2020 [48]	PLA-DX-PEG	Hydrogel
Hokugo et al., 2005 [94]	Gelatin	Hydrogel
Re et al., 2019 [95]	Chitosan	Hydrogel
Takeda et al., 2005 [96]	Collagen	Sponge made by freeze-drying
Takahashi et al., 2005 [97]	Gelatin/βTCP	Sponge made by freeze-drying
Lee et al., 2011 [100]	Collagen	Collagen nanofiber made by electrospinning, hardened with PCL
Enayati et al., 2018 [84]	PVA/HA	Fiber made by electrospinning
Deng et al., 2019 [101]	PLGA/HA/chitosan	3D printing
Zhang et al., 2019 [102]	PLGA/βTCP/GO	3D printing

PLA-DX-PEG: poly lactic acid-p-dioxanone-polyethylene glycol block copolymer, TCP: tricalcium phosphate, PCL: polycaprolactone, PVA: poly (vinyl alcohol), HA: hydroxyapatite, PLGA: polylactic-co-glycolic acid, GO: graphene oxide.

**Table 4 bioengineering-11-00180-t004:** Cells and signaling molecules used in bone-regeneration therapy.

Author, Year	Materials	Structure
Deng et al., 2019 [101]	PLGA/HA/chitosan	BMP-2
Das et al., 2016 [103]	PLAGA	BMP-6, VEGF
Berner et al., 2012 [104]	PCL	BMP-7, PRP
Liu et al., 2013 [61]	Chitosan/HA	BMSC
Cheng et al., 2018 [109]	Silk fibroin/PCL	PRP

PLGA: polylactic-co-glycolic acid, HA: hydroxyapatite, BMP: bone morphogenetic protein, PLAGA: poly(lactic-co-glycolic acid), VEGF: vascular endothelial growth factor, PCL: polycaprolactone, BMSC: bone marrow-derived mesenchymal stem cell, PRP: platelet-rich plasma.

## Data Availability

Our research data cannot be shared due to the need to protect patients privacy.

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
