# Peer review of "Bone-Regeneration Therapy Using Biodegradable Scaffolds: Calcium Phosphate Bioceramics and Biodegradable Polymers"

_bioengineering, 2024, doi:10.3390/bioengineering11020180_

Round 1
Reviewer 1 Report
Comments and Suggestions for Authors
This review manuscript is well organized and summarized in the field of ceramic and polymer bone repair materials.
One problem, in part of 3.1. Material of the Polymer, suggesting that pure polymer and polymer/ceramic should be included in detail. It is better to discuss in two sub-parts of pure polymer and polymer/ceramic. Now for some polymers such as PLA, there is no polymer/ceramic composite in detail. Please supplement.
Author Response
Point: One problem, in part of 3.1. Material of the Polymer, suggesting that pure polymer and polymer/ceramic should be included in detail. It is better to discuss in two sub-parts of pure polymer and polymer/ceramic. Now for some polymers such as PLA, there is no polymer/ceramic composite in detail. Please supplement.
Response 1: We would like to thank Reviewer 1 for providing us with helpful suggestions to improve our manuscript. Per suggestion, we created a new section entitled "3.2. Composite of Biodegradable Polymer and Calcium Phosphate Bioceramics" to provide additional details on polymer/ceramic composites.
Reviewer 2 Report
Comments and Suggestions for Authors
The scientific review on the topic Bone regeneration therapy using biodegradable scaffolds: calcium phosphate bioceramics and biodegradable polymers dedicated to materials and methods for designing composites for restoring bones in cases of extensive bone loss, as well as methods of their use in surgery. The review has both strengths and weaknesses. For example, the review has a language that is easy and understandable for a wide range of readers, the material is presented consistently and structured, and specific examples of successful use of calcium phosphate bioceramics and biodegradable polymers in complex surgical cases are shown. English is at a high level.
However, there are a number of disadvantages.
1. The review is quite condensed and includes 93 sources, some of which are more than 20 years old.
2. It would be good to summarize the literature sources of sections 3.1-3.3 into appropriate tables.
3. One of the interesting features and uniqueness of such a review may be the analysis of hydroxyapatite concentrations in composites. Could the authors note their meanings in section 3.1.
4. The review touches on the topic of adverse effects of the use of composite materials for the restoration of bone areas. Are there such statistics?
5. The review mentions the topic of degradation of biopolymers included in composites. Could the authors analyze such a parameter, and, accordingly, the effectiveness of using such materials in section 3.1.
6. The review focuses mainly on hydroxyapatite, but the name also implies other types of calcium phosphate bioceramics. Please expand on this issue a little in the review.
Author Response
Point 1: The review is quite condensed and includes 93 sources, some of which are more than 20 years old.
Response 1: We would like to thank Reviewer 2 for providing us with insightful comments and suggestions. Although we agree that some of our references are old, we believe that these references include important research contributions for the development of scaffold research. However, we also updated references related to new research in the revised manuscript according to this suggestion.
Point 2: It would be good to summarize the literature sources of sections 3.1-3.3 into appropriate tables.
Response 2: Per suggestion, we summarized our sources in sections 3.1-3.4 into a table.
Point 3: One of the interesting features and uniqueness of such a review may be the analysis of hydroxyapatite concentrations in composites. Could the authors note their meanings in section 3.1.
Response 3: We would like to thank Reviewer 2 for this insightful comment. We created a new section entitled "3.2. Composite of Biodegradable Polymer and Calcium Phosphate Bioceramics." In this new section, we cited literature that evaluated the effects of HA concentration within composite scaffolds and discussed the effects of HA concentration on the physical and osteogenic properties of the scaffold.
Point 4: The review touches on the topic of adverse effects of the use of composite materials for the restoration of bone areas. Are there such statistics?.
Response 4: The paper by Sotome et al. (Reference No. 〇〇) describes the adverse effects of HA/collagen scaffold, and performs statistical analysis. We have cited the paper to describe these adverse effects.
Point 5: The review mentions the topic of degradation of biopolymers included in composites. Could the authors analyze such a parameter, and, accordingly, the of using such materials in section 3.1.
Response 5: Per suggestion, we have added a description in section 3.1 to explain the effectiveness of degradation in the use of these materials.
Point 6: The review focuses mainly on hydroxyapatite, but the name also implies other types of calcium phosphate bioceramics. Please expand on this issue a little in the review.
Response 6: The opening sentence of section 2 has been revised to emphasize the fact that calcium phosphate bioceramics include other types such as hydroxyapatite, βTCP, and carbonate apatite.
Reviewer 3 Report
Comments and Suggestions for Authors
Dear,
Authors investigated various biodegradable polymers that are being considered for use as scaffolds for regenerative medicine are also being considered for use as synthetic bones.
- Abstract should be written and explain more
-Introduction is ok
-Method is well-explained
-Results and discussion are ok.
Best wishes,
Author Response
Point: Abstract should be written and explain more.
Response: We would like to thank Reviewer 2 for providing us with insightful comments and suggestions. As per suggestion, we have revised our abstract to provide additional details of our review article.
Reviewer 4 Report
Comments and Suggestions for Authors
The present paper is devoted to recent achievements in bone regeneration via the introduction of different types of bone substitute materials. The main classes of materials discussed are calcium phosphate ceramics and composites and polymers. The paper is not scientifically sound. The references in the paper are not up to date, the most recent cited papers were published in 2020 (the only two out of 93), some cited papers were published in the early 2000s (for example in 2007 and 2008 years).
The calcium phosphates are represented by the ceramics hydroxyapatite (HA) and tricalcium phosphate (TCP). At the same time, one of the most promising bone substitutes is octacalcium phosphate (OCP). There are many papers devoted to the compatible study of the in vivo behavior of OCP, HA and another calcium phosphate, including β-tricalcium phosphate (Kamakura et al., 2002; Suzuki et al., 1991, 2020). These publications showed that OCP is more resorbable than HA and β-tricalcium phosphate, and also promotes bone formation more than other implanted calcium phosphates. In addition, OCP biodegrades by direct resorption by osteoclast-like cells, enhancing replacement with newly formed bone during progressive implantation periods (Fuji et al., 2009; Honda et al., 2009; Kamakura et al., 1997) enhancing the replacement with newly formed bone throughout progressive implantation periods. Synthetic OCP could be considered as a bone scaffold and it is more resorbable than known HA and beta-tricalcium phosphate and also promotes bone formation more than other implanted calcium phosphates(Hayashi et al., 2022; Kamakura et al., 2002; Suzuki et al., 2020)”.
At the same time, OCP was not mentioned at all in the paper.
Fuji, T., Anada, T., Honda, Y., Shiwaku, Y., Koike, H., Kamakura, S., Sasaki, K., & Suzuki, O. (2009). Octacalcium Phosphate–Precipitated Alginate Scaffold for Bone Regeneration. Https://Home.Liebertpub.Com/Tea, 15(11), 3525–3535. https://doi.org/10.1089/TEN.TEA.2009.0048
Hayashi, K., Yanagisawa, T., Kishida, R., & Ishikawa, K. (2022). Effects of Scaffold Shape on Bone Regeneration: Tiny Shape Differences Affect the Entire System. ACS Nano, 16(8), 11755–11768. https://doi.org/10.1021/ACSNANO.2C03776/SUPPL_FILE/NN2C03776_SI_001.PDF
Honda, Y., Anada, T., Kamakura, S., Morimoto, S., Kuriyagawa, T., & Suzuki, O. (2009). The Effect of Microstructure of Octacalcium Phosphate on the Bone Regenerative Property. Https://Home.Liebertpub.Com/Tea, 15(8), 1965–1973. https://doi.org/10.1089/TEN.TEA.2008.0300
Kamakura, S., Sasano, Y., Homma-Ohki, H., Nakamura, M., Suzuki, O., Kagayama, M., & Motegi, K. (1997). Multinucleated giant cells recruited by implantation of octacalcium phosphate (OCP) in rat bone marrow share ultrastructural characteristics with osteoclasts. Journal of Electron Microscopy, 46(5), 397–403. https://doi.org/10.1093/OXFORDJOURNALS.JMICRO.A023535
Kamakura, S., Sasano, Y., Shimizu, T., Hatori, K., Suzuki, O., Kagayama, M., & Motegi, K. (2002). Implanted octacalcium phosphate is more resorbable than β-tricalcium phosphate and hydroxyapatite. Journal of Biomedical Materials Research, 59(1), 29–34. https://doi.org/10.1002/JBM.1213
Suzuki, O., Miyasaka, Y., Sakurai, M., Nakamura, M., & Kagayama, M. (1991). Bone Formation on Synthetic Precursors of Hydroxyapatite. The Tohoku Journal of Experimental Medicine, 164(1), 37–50. https://doi.org/10.1620/TJEM.164.37
Suzuki, O., Shiwaku, Y., & Hamai, R. (2020). Octacalcium phosphate bone substitute materials: Comparison between properties of biomaterials and other calcium phosphate materials. In Dental Materials Journal (Vol. 39, Issue 2, pp. 187–199). Japanese Society for Dental Materials and Devices. https://doi.org/10.4012/dmj.2020-001
Author Response
Point 1: The present paper is devoted to recent achievements in bone regeneration via the introduction of different types of bone substitute materials. The main classes of materials discussed are calcium phosphate ceramics and composites and polymers. The paper is not scientifically sound. The references in the paper are not up to date, the most recent cited papers were published in 2020 (the only two out of 93), some cited papers were published in the early 2000s (for example in 2007 and 2008 years).
Response 1: We would like to thank Reviewer 4 for providing us with insightful comments and suggestions. Although we agree that some of our references are old, we believe that these references include important research contributions for the development of scaffold research. However, we also updated references related to new research in the revised manuscript according to this suggestion.
Point 2: The calcium phosphates are represented by the ceramics hydroxyapatite (HA) and tricalcium phosphate (TCP). At the same time, one of the most promising bone substitutes is octacalcium phosphate (OCP). There are many papers devoted to the compatible study of the in vivo behavior of OCP, HA and another calcium phosphate, including β-tricalcium phosphate (Kamakura et al., 2002; Suzuki et al., 1991, 2020). These publications showed that OCP is more resorbable than HA and β-tricalcium phosphate, and also promotes bone formation more than other implanted calcium phosphates. In addition, OCP biodegrades by direct resorption by osteoclast-like cells, enhancing replacement with newly formed bone during progressive implantation periods (Fuji et al., 2009; Honda et al., 2009; Kamakura et al., 1997) enhancing the replacement with newly formed bone throughout progressive implantation periods. Synthetic OCP could be considered as a bone scaffold and it is more resorbable than known HA and beta-tricalcium phosphate and also promotes bone formation more than other implanted calcium phosphates(Hayashi et al., 2022; Kamakura et al., 2002; Suzuki et al., 2020)”. At the same time, OCP was not mentioned at all in the paper.
Response 2: We would like to thank Reviewer 4 for this detailed comment. Per suggestion, we added descriptions of carbonate apatite and OCP in section 2, and OCP composites in section 3.2, and cited the relevant literature that Reviewer 4 introduced to us.
Reviewer 5 Report
Comments and Suggestions for Authors
For a review it is acceptable for publication. The manuscript covers the relevant literature data. Indeed the calcium phosphate can be a solution for synthetic bones but nobody wish it …Although titanium is also not a solution.
Author Response
Point: For a review it is acceptable for publication. The manuscript covers the relevant literature data. Indeed the calcium phosphate can be a solution for synthetic bones but nobody wish it …Although titanium is also not a solution.
Response: We would like to thank Reviewer 5 for providing a favorable response to our manuscript.
Round 2
Reviewer 4 Report
Comments and Suggestions for Authors
The authors improved the manuscript. The paper could be accpted.